# Towards a better understanding of risk selection in maternal and newborn care: A systematic scoping review

Bahareh Goodarzi[1]*, Annika Walker[1], Lianne Holten[1], Linda Schoonmade[2], Pim Teunissen[3,4], François Schellevis[5,6], Ank de Jonge[1]

1 Department of Midwifery Science, AVAG, Amsterdam Public Health Research Institute, Amsterdam UMC, Vrije Universiteit Amsterdam, Amsterdam, The Netherlands, 2 Medical Library, Vrije Universiteit Amsterdam, Amsterdam, The Netherlands, 3 Department of Obstetrics and Gynaecology, Amsterdam UMC, Vrije Universiteit Amsterdam, Amsterdam, The Netherlands, 4 School of Health Professions Education, Maastricht University, Maastricht, The Netherlands, 5 Department of General Practice and Elderly Care Medicine, Amsterdam Public Health research institute, Amsterdam UMC, Vrije Universiteit Amsterdam, Amsterdam, The Netherlands, 6 NIVEL (Netherlands Institute for Health Services Research), Utrecht, The Netherlands

* b.goodarzi@amsterdamumc.nl

**Data Availability Statement:** All relevant data are within the manuscript and its Supporting Information files.

## Abstract

Globally, millions of women and their children suffer due to preventable morbidity and mortality, associated with both underuse and overuse of maternal and newborn care. An effective system of risk selection that differentiates between what care should be provided and who should provide it is a global necessity to ensure women and children receive appropriate care, at the right place and the right time. Poor conceptualization of risk selection impedes evaluation and comparison of models of risk selection across various settings, which is necessary to improve maternal and newborn care. We conducted a scoping review to enhance the understanding of risk selection in maternal and newborn care. We included 210 papers, published over the past four decades, originating from 24 countries. Using inductive thematic analysis, we identified three main dimensions of risk selection: (1) risk selection as an *organisational measure* to optimally align women's and children's needs and resources, (2) risk selection as a *practice* to detect and assess risk and to make decisions about the delivery of care, and (3) risk selection as a *tool* to ensure safe care. We found that these three dimensions have three themes in common: risk selection (1) is viewed as both *requiring and providing regulation*, (2) has a *provider centred focus* and (3) aims to *avoid underuse of care*. Due to the methodological challenges of contextual diversity, the concept of risk selection needs clear indicators that capture the complexity of care to make cross-setting evaluation and comparison of risk selection possible. Moreover, a comprehensive understanding of risk selection needs to consider access disparity, women's needs, and unnecessary medicalization.

## Introduction

Maternal and newborn care (hereafter, MNC) services fail to meet many of the essential needs of childbearing women (when we use the term 'woman', we also refer to individuals with a

**Funding:** The author(s) received no specific funding for this work.

**Competing interests:** The authors have declared that no competing interests exist.

uterus who are not woman identified, including trans men and non-binary individuals) and their unborn or newborn children (hereafter, children). Every year there are an estimated 139 million births worldwide [1]. An estimated 303,000 women die during pregnancy, birth or soon after [2], 2.6 million babies are stillborn [3] and 2.5 million infants die in the first month of life [4]. Maternal and perinatal morbidity and mortality are partly preventable. In the USA, for example, an estimated 63% of pregnancy-related maternal deaths [5] and 27%-54% of infant deaths among children born full-term are avoidable [6]. The Lancet series addressing maternal and newborn health [7–12] emphasized that preventable morbidity and mortality is associated with both underuse and overuse of care. A meta-synthesis in the Lancet series on Midwifery [12] pointed out that childbearing women and their children need a care system that helps them to stay healthy, and that provides a timely transition to medical specialist care for those who develop complications.

Pregnancy and birth are primarily physiological processes but risks and complications can occur. The degree and type of risk related to pregnancy and birth differs between women and children, and between countries and care settings. To ensure women and children receive appropriate care, at the right place and the right time, an effective system of risk selection that differentiates between what care should be provided and who should provide it is a global necessity [13,14].

Kennedy and colleagues [15,16] identified the evaluation and comparison of models of risk selection across various settings as one of the top research priorities necessary to improve care. However, a key problem impeding cross-setting evaluation and comparison rests on the poor conceptualization of risk selection. The lack of conceptual clarity hinders the development of an evidence base for the most effective strategies to organise and practice risk selection. This is illustrated by the variety of ways in which risk selection is operationalized in research. For example, some scholars see risk selection as a skill in terms of health care professionals' cognitive process [17–19]. Others understand risk selection as a means of organizing care [20–22]. Sometimes, risk selection is defined as a tool, for example in the application of risk indicators [23,24], the use of guidelines and protocols [25,26], and screening instruments [27–29]. Often, risk selection is seen as a safety system, meant to minimize morbidity and mortality mainly due to undertreatment [30–32]. A comprehensive understanding of risk selection, encompassing the relationship between these operationalisations remains absent, indicating a lack of shared conceptualisation of what risk selection entails in MNC.

To enhance the understanding of risk selection in MNC we conducted a scoping review. We systematically searched the scientific literature, and examined papers spanning the last four decades to identify key dimensions of risk selection, using the following research question: how is the selection of childbearing women and children that require specialized care because of increased medical risks or actual complications conceptualized?

## Methods

We conducted a scoping review, using a systematic design for the search and data selection, and inductive thematic analysis for the data analysis and data synthesis. We used a scoping review methodology based on the framework outlined by Arksy & O'Malley [33]. To enhance the framework, we took into consideration the following recommendations by Levac and colleagues [34] and Daudt and colleagues [35]: (1) we conducted considerable research about review studies to ensure an appropriate match between our research interest and the methodology, (2) we articulated a clear research question, rationale and purpose of the scoping review, which led the decision making throughout the study, (3) we assembled a research team with content and methodological expertise, consisting of an information specialist and researchers

from the fields of general medical practice, obstetrics, midwifery, anthropology, and psychology, (4) two reviewers conducted the selection of publications, (5) we used a charting form and qualitative content analysis approach for the data extraction, and (6) we report the results and consider the meaning of the findings as they relate to the purpose of the study and research question. The study protocol was not registered.

## Search strategy

The literature search was conducted in accordance with the Preferred Reporting Items for Systematic Reviews and Meta-Analysis (PRISMA) statement extension for scoping reviews [36] (S1 Table). A comprehensive search was performed in the bibliographic databases PubMed, Embase.com, Cinahl (via Ebsco) and the Cochrane Library, in collaboration with a medical librarian (LS). Search terms included controlled terms (MesH in PubMed, Emtree in Embase and Cinahl Headings). We used free text term only in the Cochrane database. The search was conducted from inception to April 16[th] 2019.

Considering the breadth of the subject, we limited the scope of our review to risk selection based on medical risk factors, excluding risk selection based on social risk factors. Hence, the following terms, including synonyms and closely related words, were used as index terms or free-text words: "risk", "selection", "maternal and newborn care", and "quality of care". The search was performed without date or language restrictions. Duplicate articles were excluded. The search results were imported and merged in the reference database Mendeley [37]. The full search strategies for all databases can be found in S2 Table.

## Selection criteria, data extraction and analysis

We used a systematic two-stage screening process to assess the relevance of the papers identified in the search [38]. In the first stage, two researchers (BG and AW) independently screened the papers' title and abstract for inclusion. To ensure inter-reviewer agreement, BG and AW met weekly to discuss uncertainties, and they specified and expanded the inclusion and exclusion criteria. In the second stage, the papers' full texts were assessed for eligibility. To ensure reviewer agreement, BG and AW assessed the first 20 full-texts independently, which resulted in complete agreement on inclusion. BG assessed the remaining papers by herself. A search update was conducted. AW randomly screened 500 of the additionally identified papers' title and abstract independently and BG and AW assessed the first 15 papers' full-texts independently, which reconfirmed inter-reviewer agreement. BG assessed the remaining title and abstracts and full-texts of the papers identified in the search update by herself.

We excluded papers published prior to the year 1981, and non-research papers such as statements, opinions, book chapters, guidelines, protocols, conference posters and presentations to enhance feasibility. We restricted the language to English and Dutch. Studies conducted in low income countries and war zones were excluded to enhance comparability.

Only studies were included focusing on the relation between the selection of medical risks, and referral between medical specialists delivering MNC. An inductive thematic synthesis approach [34,39,40] was used to identify how the concept of risk selection was approached in the included papers. Because we were interested in the operationalization of risk selection, we focused on the background, methods and discussion sections of the papers, and we did not assess the study results and the methodological quality of the papers. Following in-depth reading, we mapped the main focus of each paper using an excel sheet, which we then organised into main categories. We then searched for overarching themes (S3 Table). Data saturation was reached after data extraction of 125 papers. Screening of the remaining papers did not result in new main categories, confirming data saturation.

## Results

Our systematic search resulted in a total of 8,509 references. Following evaluation of their title and abstract, 371 papers remained for full text review. After reviewing these papers, 171 papers did not meet our inclusion criteria, leaving 210 papers for analysis (Fig 1). The papers' study objective and study design are shown in S4 Table. The majority of the included papers used quantitative methods. We found an increase in the number of papers published over the past four decades, with a peak in the years 1989 (n = 9) and 1995 (n = 9) and more than half of the papers published in the last 10 years (2009–2018). The included papers originated from 24 high income countries. Most papers originated from North America, Europe and Australia, with the highest numbers of papers from the USA (n = 55), the Netherlands (n = 48), England (n = 27) and Australia (n = 27).

We identified three main categories, which can be seen as the three dimensions of risk selection; (1) risk selection as an *organisational measure* to optimally align women's and children's needs and resources, (2) risk selection as a *practice* to detect and assess risk and to make decisions about the delivery of care, and (3) risk selection as *a tool* to ensure safe care. We found that these dimensions of risk selection had three themes in common: across these dimensions risk selection is (1) viewed as both *requiring and providing regulation*, (2) has a *provider centred focus* and (3) aims to *avoid underuse of care* (Fig 2). In all papers one or more main categories were present. In S4 Table we show the dominant category per paper. We illustrate our findings by referring to the most relevant papers.

### Risk selection as an organisational measure: Aligning risk and resources

Papers focussing on organisation of MNC refer to risk selection as a means for care systems to manage a common challenge: meeting the needs of childbearing women and their children with limited resources. These needs are referred to in terms of 'risk'. Risk has a negative connotation, associated with pathology or abnormality, and is described using terms such as, 'illness' [42], 'complication' [43], 'disease', [44], 'problem' [21], 'disorder' [45]. In the included papers, risk sometimes indicates an *unwanted event* [46,47], the *cause of an unwanted event* [48,49], or the *probability of an unwanted event* [50,51], which may or may not occur, adversely effecting outcomes of care.

Risk selection is used to allocate resources and align women's and children's needs with MNC services. Allocation of resources is considered effective when necessary care is provided by health care professionals with the appropriate level of expertise, in the most appropriate place, where the appropriate facilities and resources are located, with the type and timing of care planned appropriately [52–57]. For example, Posthumus and colleagues [58] studied the interaction between access to care, care demand and care supply. According to Reddy and colleagues [59] effective use of resources implies that specialist services should be reserved for women with complications or those at most risk of developing complications. Nuovo [60] pointed out that risk selection is especially important when consulting care providers do not have immediate access to specialist care, enabling them to plan the necessary care.

In the papers addressing the organisation of MNC, at least one of the following attributes of healthcare services is discussed as precondition to optimally align needs and resources: availability, access, and timeliness of care [61]. Risk selection is described as an instrument to balance access to, and availability of resources with the perceived likelihood of needs, while ensuring timeliness of care. Timeliness refers to the geographical distribution of resources, and the distance and time to reach them. As such, transportation is emphasized as essential to achieve timely care [62–66]. Availability of resources includes availability of expertise, facility, technology, diagnostics and therapy [67–74]. For example, Hein & Burmeister [75] explained

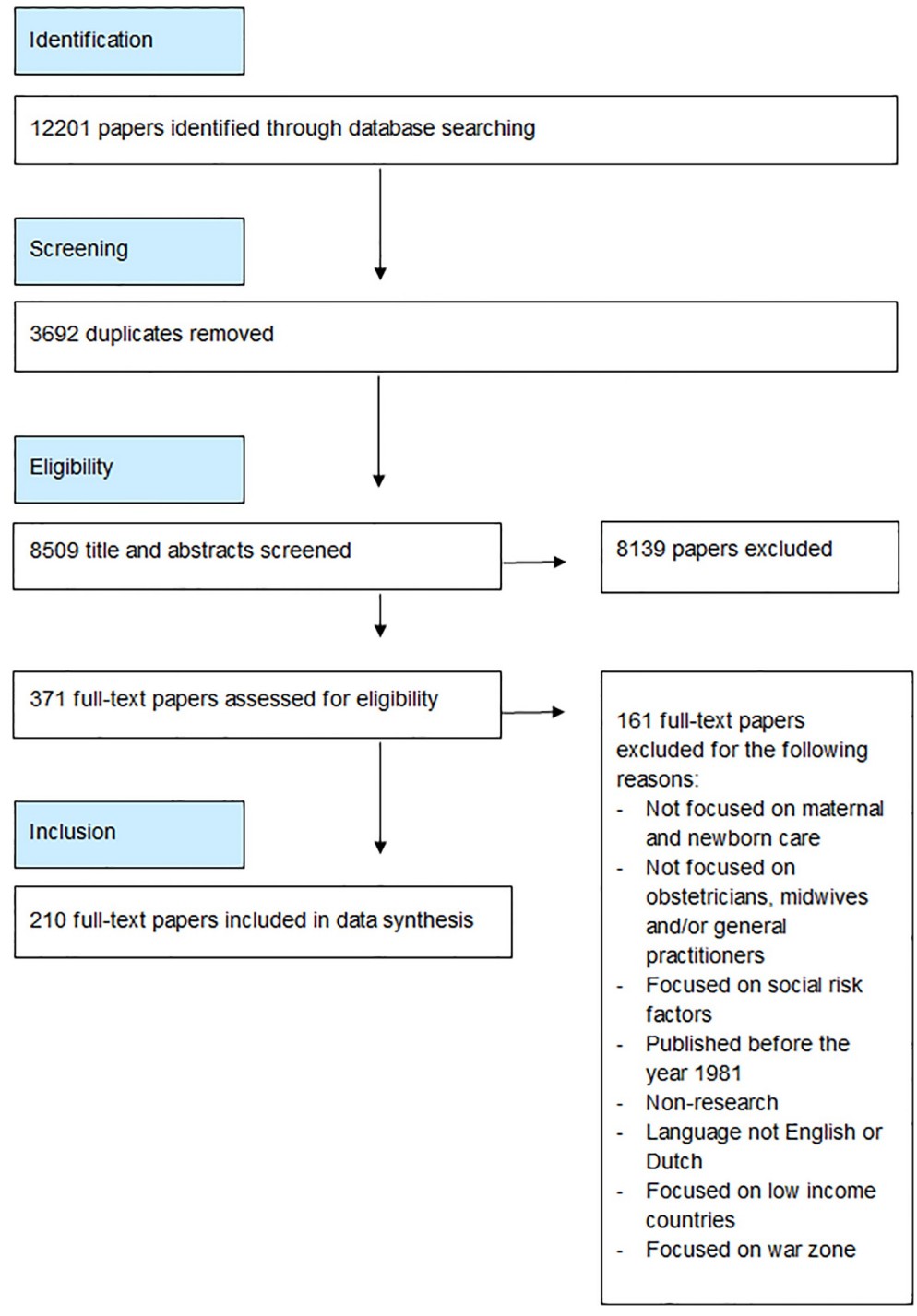

**Fig 1. Study selection process [41].**

that risk selection enables care providers to anticipate the limitations of their own facilities, and Smit and colleagues [72], studied whether access to pulse oximetry for the assessment of infants born in a community based midwifery care setting could prevent referral to a paediatrician. Our analysis also showed that regulation of access to resources is sometimes used to

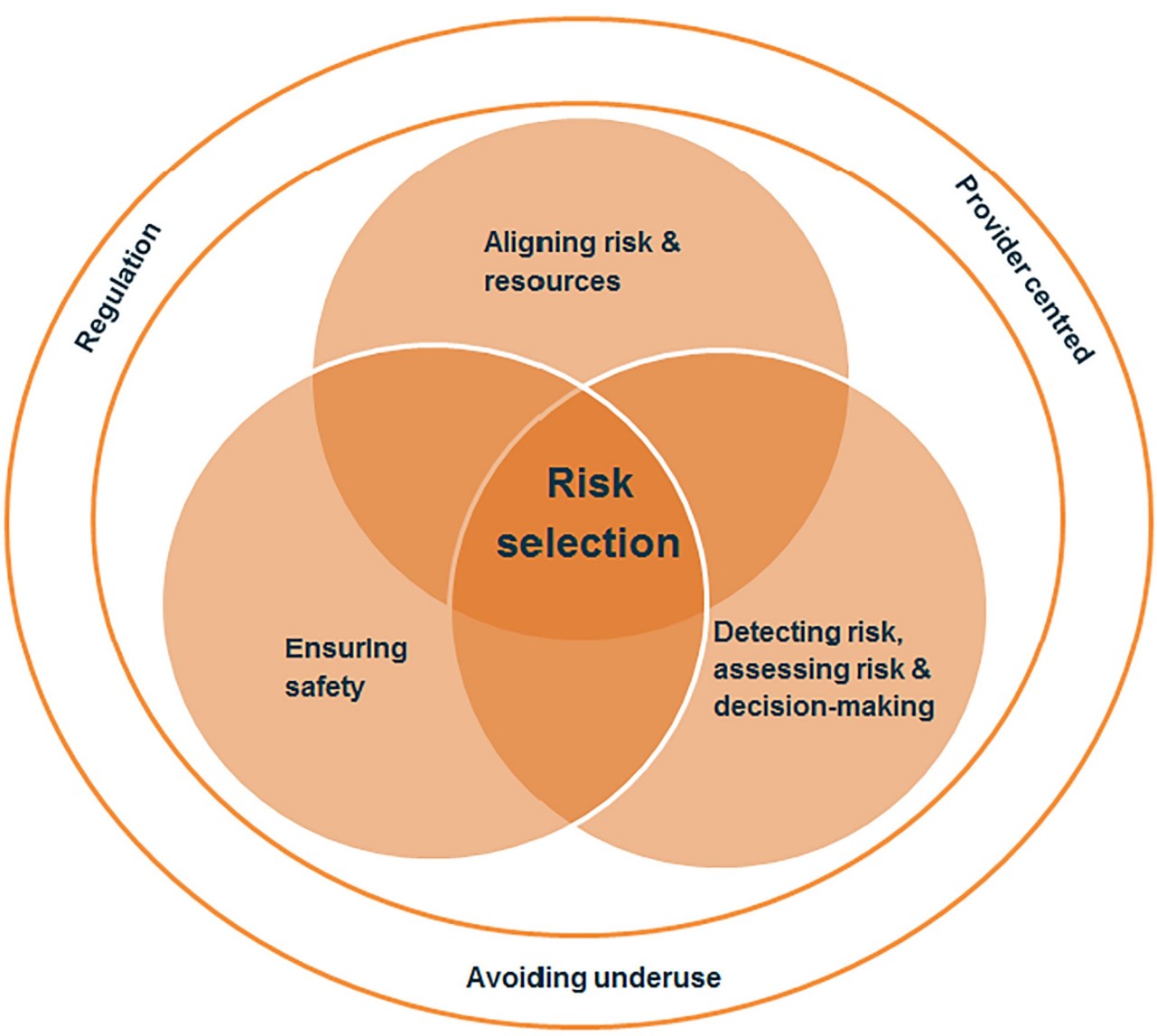

**Fig 2. Identified dimensions and shared themes.**

optimally align needs and resources. We found three main approaches of access regulation; geographical, medical and financial. Geographical regulation is based on the location of women and children and care facilities. Regionalization is the most applied strategy of care access regulation, organizing care in different levels, by geographically spreading general services at the lowest level of care, and concentrating scarce resources and specialized services at the highest level of care [57,73,76–79]. Medical regulation is grounded in women's and children's needs, expressed in terms of medical risk criteria; care is provided only to those whose needs correspond to prevailing criteria [59,80–85]. In financially regulated access to care, remuneration of care provision is subject to certain requirements and restrictions. Insurers apply a variety of strategies, such as capitation schemes and mandatory authorization for treatment decisions and referral [86,87]. In some countries, care access is medically regulated,

where women and children can only access medical specialists via a primary care provider, sometimes referred to as 'gatekeeper' [86,88,89]. Britt and colleagues pointed out that many insurers used ". . .primary-care physicians to act as gatekeepers who must approve referrals to specialists and sub-specialists. . ." [90] ". . .to limit the use of so-called 'unnecessary' referrals. . ." [86] and ". . .keep costs down. . ." [90]. In their study about referrals for genetic counselling by GPs in the Netherlands, Aalfs and colleagues [91] explained that ". . .as stated in an agreement between the Dutch Society of Clinical Genetics and the Dutch health insurance companies, referral for genetic counselling to one of the eight academic centres for clinical genetics was the task of GPs exclusively. This means that every patient who wants to be referred for genetic counselling has to visit their GP first."

## Risk selection as a practice: Detecting and assessing risk, and making decisions about the delivery of care

Part of the included papers address risk selection as a practice of detecting and assessing risk and making decisions about the delivery of care. Overall, this process is perceived in two contrary ways; on the one hand risk selection is defined as an objective and straightforward process; on the other hand risk selection is viewed as a subjective and complex process. As an objective process, risk is considered predictable and detectable using many data sources, including screening and diagnostic methods, such as risk scoring [60,92], a partogram [28], fetal fibronectin bedside testing for diagnosing preterm labour [27], and abdominal palpation and ultrasound for determining fetal presentation [93] and fetal growth [94]. In their paper about obstetrician-gynaecologists' management of mental health conditions, Leddy and colleagues [95] explained that "the purpose of screening is not to determine that complete realm of psychological needs of a patient, but instead is a means by which to identify patients who may require further assessment, monitoring or referral." Many papers predominantly consider risk selection as a process with a dichotomous outcome, risk classified as either present or absent. Detected risks can relate to the mother and the child, and differ in their nature, severity and urgency [65,66,96,97]. As a subjective process, authors acknowledge risk selection's complexity, determined by health care organisation, care providers and women [98]. Organisation characteristics include the number of care providers involved in care provision, location, communication, collaboration, and geography [30,58,69,93,99–113]. Health care providers' perceived risk, knowledge, expertise, confidence, personal views, awareness and attitude, financial considerations and women's characteristics and preference, amongst others, are described as decisive factors [17–19,26,46,86–88,95,99,100,105,107,109,114–118]. Providers' behaviour is considered influenceable via, for example, education [97,119–121], and guidelines [19,97,115,122]. Women bring in factors such as timing of presentation, sense of control, views and beliefs, sense of safety, perceived norms, perceived availability of options, and demographic characteristics such as age, level of educational and income [114,115,123–130]. Because a consulting care provider either needs additional resources or does not, the decision to refer is inherently a "threshold phenomenon" [108]. The contributing factors in the decision making process are weighed differently by different care providers, resulting in varying referral-thresholds, thus practice variation. According to several papers [19,63,108,117,131], this is especially the case for intermediate levels of risk, also referred to as the "grey zone". In contrast to clear high or low risks, these risks ". . .may be near the referral threshold, and therefore disproportionately susceptible to the marginal influences of numerous personal, social, cultural, and financial considerations. . ." [108].

   The practice of risk selection in terms of detecting and assessing risk is not reserved to a certain profession, but rather performed by all professionals involved in care provision. The

criteria, policy, procedures and division of professional tasks and responsibilities, are laid down in local and international, monodisciplinary and multidisciplinary agreements, defining women's and children's needs for healthcare services accordingly [26,51,59,84,85,115,120,122,132–136].

Timing is considered a highly important feature of the detection, assessment and decision making process. Generally, risk selection is addressed in two ways; the time until risk detection [28,31,137–139] and the time between risk detection and care provision [30,31,66,101,140,141]. It is emphasized that prolonged time between risk detection and care provision can cause delay, leading to preventable morbidity and mortality.

## Risk selection as a tool: Ensuring safety

The majority of the papers we reviewed perceived risk selection as a tool to ensure safe care, the shared notion being that adequate risk selection results in safe care. Risk selection as a tool to ensure safe care is regulated by designated bodies [30,75,104,139,142]. The quality of risk selection is considered measurable, reflected by care outcomes. Authors assess the quality of risk selection predominantly by short-term and quantitative outcome measures, most frequently using referral rates, intervention rates and morbidity and mortality rates. The premise is that high quality risk selection results in low morbidity and mortality rates, and cases of maternal and perinatal morbidity and mortality that occur in specialist care reflect providers' ability to screen for risks, to make a correct diagnosis, and to refer timely [30,31,62,101,140,143–148]. For example, Ferrazzi and colleagues [85] reflected on the results of their study on the outcomes of midwife-led labour in low-risk women: "as expected, maternal outcomes, such as mode of delivery, episiotomy and PPH, were significantly higher in women with compared to those without emerging risks identified by midwives. On the other hand, fetal outcomes were not significantly different between the two groups. This might be interpreted as a consequence of the quality of midwives' assessment of risk during labor, which allowed for early diagnosis and prompt treatment of incoming complications".

While most studies focus on risk selection as a tool to ensure safe care, the standards for optimal risk selection and the optimality threshold remain unclear. Care outcomes are interpreted through cross-setting comparison, using a variety of reference points, including population, location and practice. For example, Rowe and colleagues [131] compared different types of maternity units, Fullerton and colleagues [149] compared local and national outcomes, Law and colleagues [150] compared midwife managed and obstetrician managed care, Romijn and colleagues [117] compared primary care midwives, clinical midwives and obstetricians, and Blondel and colleagues [151] compared countries. Furthermore, for measuring the quality of risk selection, quality is defined in various ways. A clear example is the diverse use of referral rates as a quality measure. According to Blix and colleague [65], "transfers should not be regarded as an adverse outcome, and are not necessarily indicators of quality of care". Across the papers, high referral rates are perceived as indicators for effective risk selection [127,152] as well as failing risk selection [17,25,132,133]. Low referral rates are perceived in the same way; as indicators for effective risk selection [127] and failing risk selection [122,153]. Fourteen of the 210 papers use women's experience as outcome of risk selection [61,124–126,128–130,154–160] and one paper included partners' experience as quality indicator for risk selection [161].

## Overarching themes: Regulation, provider centred focus and avoiding underuse

We found that the three dimensions of risk selection—an organisational measure to allocate resources, a practice to detect and assess risk and to make decisions about the delivery of care,

and as a tool to ensure safe care—have three themes in common: regulation, provider centred focus and avoiding underuse of care. First, risk selection is viewed as both requiring *and* providing regulation. Risk selection regulates allocation of tasks and responsibilities based on geographical, medical and/or financial criteria. In terms of risk detection and assessment and decision making about the delivery of care, risk selection regulates delivery of care, and is supported by local and international, monodisciplinary and multidisciplinary agreements. Risk selection as a tool to ensure safe care determines what is considered safe and is regulated by designated bodies. The second commonality is the provider centred focus; a small minority of the papers addressed women's experiences. At the organisational and practice level women and children gain access to care only after risk is detected or confirmed by the provider. At the control level, outcome is predominantly viewed and evaluated as providers' conduct. Thirdly, aligning needs and resources, assessing, detecting and deciding on risks, and risk selection as a means to ensure safe care share the focus of preventing underuse of care.

## Discussion

We conducted this review to enhance the conceptual understanding of risk selection in MNC, which is necessary to optimise the organisation and practice of risk selection in MNC. The included 210 publications from 24 high income countries indicate that risk selection is a universal phenomenon, used to differentiate between what care should be provided and who should provide it to ensure women and children receive appropriate care, at the right place and the right time [13,14]. Thematic inductive synthesis identified three main dimensions of risk selection. (1) Risk selection is used at the level of health care organisation as a means to allocate resources, align women's and children's needs with healthcare services, and balance access to, and availability of resources with the perceived likelihood of needs, while ensuring timeliness of care. (2) At the practice level, risk selection refers to detecting and assessing risk and making decisions about delivery of care. (3) Risk selection is also used as a tool to ensure safe care. We found that these three dimensions have three themes in common: risk selection (1) is viewed as both requiring and providing regulation, (2) has a provider centred focus, and (3) aims to avoid underuse of care (Fig 1).

### Evaluating the quality of risk selection: The challenge of contextual diversity

Our results show a paradox in the understanding of risk selection. On the one hand, risk selection is often assessed by cross-setting comparison, using quantitative, short-term, and infant outcome measures. On the other hand, our results emphasize the complexity of risk selection, showing that the operationalization of risk selection is highly contextualized, determined by numerous factors including geography, demography, government policy, laws and regulations, history and culture. For example, Scherjon and colleagues [20] and Papiernik and colleagues [22] discussed how these factors influence the organisation of risk selection in different countries. Some papers call for acknowledging the ambiguous nature of risk, emphasizing the constructionist character of risk [88,117,123,133,134,155,162], and address the impossibility to detect and eliminate all risks [47,53,94,163]. According to Reddy and colleagues [59] "it should be remembered that the risk status of a woman may change during the course of pregnancy", and sometimes no measures can be found to improve care [30,31,164]. The paradoxical perception of risk selection as objective and measurable versus relative and contextual is present throughout the included papers. The complexity of risk selection, for instance, is often emphasized [17,26,31,61,88,104,109,116,117,142,165]. Nevertheless, scholars rarely consider complex

metrics such as underlying causes of care outcomes, unnecessary interventions, long-term outcomes and inter-professional collaboration.

The contextual relativity of risk selection is a major challenge for cross-setting evaluation, complicating comparability necessary for evaluating risk selection's quality [166–168]. One of the problems is establishing equal understandings of quality indicators [16,167]. This explains the absence of clear and shared standards for optimal risk selection in our findings. For example, we found that transfer rates used as a quality measure of risk selection were interpreted in different ways. The absence of standards makes it "difficult to assess what transfer rate provides the best outcomes of care" [65]. It also hinders meaningful evaluation of the quality of risk selection because, as Offerhaus and colleagues [169] pointed out, "high intrapartum referral rates suggest that some of the referrals. . . might have been unnecessary. . ., on the other hand, . . .achieving a low referral rate is no goal in itself". This ambiguity impedes recommendations to improve risk selection.

Only through careful, context-specific evaluation, with understanding of the reasons for variations, can cross-setting comparisons support the quest for quality improvement of risk selection. This calls for the use of composite measures for complex phenomena that capture the complexity of care, such as the interactions between cognitive, social and cultural factors [16]. A lack of contextual sensitivity in cross-setting comparisons may lead to misconceptions and erroneous policy decisions, leading to unsuccessful initiatives aiming to optimise risk selection. Furthermore, the 24 countries that are covered in this review are not represented evenly, and some countries are not represented at all, such as Switzerland, Croatia, Cyprus, Hungary and Luxembourg. Studies of risk selection in MNC from the underrepresented countries are necessary to gain insight in local practices, necessary for cross-setting learning.

## Blind spots of risk selection: Equitable access to care, women-centred care, and overuse of care

Risk selection aims to ensure that women and children receive appropriate care, at the right place and the right time, predominantly by pursuing efficient, timely and safe care. However, to optimize risk selection, other aspects of quality of care, namely equitable access to care, women-centred and effective care [170], require further consideration.

**Equitable access to care.**   A precondition to ensure that women and their children receive the care they need is equal opportunities to access care for those with equal needs, regardless of personal characteristics,– such as gender, age, ethnicity, geographic location and socioeconomic status [170,171]. However, our results show that care provision is determined by many characteristics—often a combination, or intersection [172] of characteristics—including characteristics of health systems and care providers [173–179]. Although some salient potential barriers to care, such as geography, are considered in the papers, the notion of equity of access to care remains unaddressed. Including equitability in risk selection will contribute to creating awareness, and prioritizing the challenge of bias and social injustice in risk selection in MNC, which is necessary to optimize risk selection.

**Women-centred care.**   Care providers hold a central position in risk selection. Not only do they co-manage access to care, risk selection is viewed and assessed as providers' conduct. The provider–woman dynamic in risk selection, and the tension that can arise when they disagree, is the focus of only one study in this review [123], and thus remains largely undiscussed. Care outcomes are measured by providers' performance indicators; of the 210 included papers, only fourteen papers encompass women's perspectives, and only one paper includes partners' experience of risk selection. Renfrew and colleagues [13] developed a women-centred framework for quality MNC, as part of the Lancet series on midwifery, showing that women highly

value communication, respect and understanding, and care that is tailored to their needs [12]. Reflecting on this framework, global health stakeholders in the Lancet's Series on Midwifery stress women's perspectives in MNC evaluation, recommending studies to "assess the views and preferences of women and families across a variety of settings about their experiences of maternal and newborn care..." [15]. The number of papers addressing women's and partners' perspectives on risk selection has increased over the years, indicating an advancing awareness in MNC about women-centred care. Further including the perspective of women in the understanding of risk selection creates room for women's individual unique needs [159,180–183], optimizing the alignment of risk selection with women's needs.

**Effective care.** The contemporary understanding of risk selection holds a blind spot for overuse of care. Our analysis shows that risk selection is focused on preventing underuse of care. In the included papers, the quality of risk is evaluated by questioning timeliness of care, in terms of whether more care delivered sooner could have led to better care outcomes [24,30–32,90,104,137,140,155]. A few papers in our review discussed the notion of a "cascade of interventions" [17,169,184], "...where one intervention in a labouring woman leads to another and so on" [17], warning for overuse of care. The vast majority of the studies focused on upscaling of care; referral from generalist tot sub-specialist care. The appropriateness of the referral, however, remained largely undiscussed. Although care that is provided "too little too late" [8] is a global problem, overuse of care is increasingly associated with poor quality care and preventable maternal and perinatal morbidity and mortality, also in high income countries [7,8,185]. Variation in care and rising rates of interventions without evident benefit [185] are indicators of care that is delivered "too much too soon" [8]. Our results indicate that this is particularly related to intermediate levels of risk, also referred to as the "grey zone" [186–195]. According to Brownlee and colleagues [185], most health care services fall into the grey zone—which include services that offer little health benefit, those for which the balance between benefits and harms differs amongst individuals, and the numerous services that are backed by little or no scientific evidence. Excluding the notion of overuse of care and downscaling of care services in the understanding of risk selection undermines the purpose of risk selection and impedes enhancing the effectivity of risk selection [8]. The challenge is finding the right balance in effective delivery of care, striving for risk selection that is not only effective in terms of preventing harm due to underuse of care, but also avoiding overuse of care.

## Strengths and limitations

To our knowledge, this is the first study to review the concept of risk selection in MNC. The search was systematically conducted with the help of an information specialist and was updated to include recent publications. We made an effort to include a wide and comprehensive range of terms in the literature search strategy. Our broad search strategy, the search update and the inductive thematic data synthesis approach enabled us to obtain a broad and deep view of the operationalisation of risk selection across the full scope of MNC during the past four decades. Due to the vast number of included papers, we restricted the search strategy to four databases that usually cover MNC literature, we only included papers in English and Dutch, and we did not apply the snowball method to extend the search. Furthermore, we only included studies conducted in high-income countries, including studies from 24 countries. Not all countries were evenly represented in our results, and some countries were not represented at all. We can only speculate on the underlying reasons, including that our search did not identify all relevant studies, for example because papers may have been published in local —non-English—journals, which are not clearly indexed within the scientific databases. However, we did reach data saturation.

## Conclusion

This systematic scoping review shows that current understanding of risk selection encompasses three main dimensions. Risk selection is used at the level of health care organisation as a means to allocate resources, align women's and children's needs with healthcare services and balance access to, and availability of resources with the perceived likelihood of needs, while ensuring timeliness of care. At the practice level, risk selection refers to detecting and assessing risk and making decisions about delivery of care. Risk selection is also used as a tool to ensure safe care. We found that these three dimensions have three themes in common; across these dimensions, risk selection is viewed as both requiring and providing regulation, has a provider centred focus, and aims to avoid underuse of care. Due to the methodological challenges of contextual diversity, the concept of risk selection needs clear indicators that capture the complexity of care to make cross-setting evaluation and comparison of risk selection possible. Moreover, a comprehensive understanding of risk selection needs to consider access disparity, women's needs, and unnecessary medicalization.

## Supporting information

**S1 Table. PRISMA checklist.**
(DOCX)

**S2 Table. Search strategies.**
(DOCX)

**S3 Table. Overarching themes, main categories and sub categories.**
(DOCX)

**S4 Table. Charactristics of included references.**
(DOCX)

**S5 Table. List of included references.**
(DOCX)

## Acknowledgments

We thank Yvonne Smit and Michelle ten Berge for their support during the first phase of this study, and Henk Koffijberg, Michelle Willard and Gülgen Öztas, for helping with retrieving the full texts.

## Author Contributions

**Conceptualization:** Bahareh Goodarzi, Pim Teunissen, François Schellevis, Ank de Jonge.

**Data curation:** Bahareh Goodarzi.

**Formal analysis:** Bahareh Goodarzi, Annika Walker, Lianne Holten, Pim Teunissen, François Schellevis, Ank de Jonge.

**Investigation:** Bahareh Goodarzi.

**Methodology:** Bahareh Goodarzi, Linda Schoonmade, Pim Teunissen, François Schellevis, Ank de Jonge.

**Project administration:** Bahareh Goodarzi.

**Supervision:** Ank de Jonge.

**Writing – original draft:** Bahareh Goodarzi.

**Writing – review & editing:** Bahareh Goodarzi, Annika Walker, Lianne Holten, Linda Schoonmade, Pim Teunissen, François Schellevis, Ank de Jonge.

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
