## [Editor Report · Decision Letter 0]

8 Jan 2020

PONE-D-19-34522

Towards a better understanding of risk selection in maternal and newborn care: a systematic scoping review

PLOS ONE

Dear Ms Goodarzi,

Thank you for submitting your manuscript to PLOS ONE. After careful consideration, we feel that it has merit but does not fully meet PLOS ONE’s publication criteria as it currently stands. Therefore, we invite you to submit a revised version of the manuscript that addresses the points raised during the review process.

Before sending for peer review, I would like to ask you to expand the information included in Supplementary File S3. Reviewers will not be able to assess how the authors have arrived at their results (the different dimensions and themes), based on the limited information included in the file S3. As a minimum, I would suggest additions to the table of characteristics of the included studies: a) a column that indicates the document type – in this way, from the scoping the reader can understand the literature included - from the inclusion and exclusion criteria we can only determine that the document are research papers – it would be good to have additional detail, for example, if it was qualitative, quantitative; and, more importantly; b) key themes and categories extracted from each paper – relating to the dimensions and themes you have highlighted. Please discuss among your team and decide what information would be useful to include in the table that would facilitate for readers a link to what was in the primary studies and your summary.  You may subsequently decide to modify your findings and discussion section, based on this additional information.

We would appreciate receiving your revised manuscript by Feb 22 2020 11:59PM. To enhance the reproducibility of your results, we recommend that if applicable you deposit your laboratory protocols in protocols.io, where a protocol can be assigned its own identifier (DOI) such that it can be cited independently in the future. For instructions see: http://journals.plos.org/plosone/s/submission-guidelines#loc-laboratory-protocols

A rebuttal letter that responds to the point raised by the Academic Editor. This letter should be uploaded as separate file and labeled 'Response to Reviewers'.A marked-up copy of your manuscript that highlights changes made to the original version. This file should be uploaded as separate file and labeled 'Revised Manuscript with Track Changes'.An unmarked version of your revised paper without tracked changes. This file should be uploaded as separate file and labeled 'Manuscript'.

We look forward to receiving your revised manuscript.

Kind regards,

Anayda Portela

Academic Editor

PLOS ONE
---

## [Author Response · Author response to Decision Letter 0]

24 Jan 2020

Dear Dr. Portela, 

Thank you for considering our manuscript for publication in PLOS ONE. We appreciate your feedback. 

We have discussed your feedback among our team and seriously considered your comments, which has resulted in revisions of the manuscript. 

We agree that additional details about the included papers and our analysis can provide the reviewers and the readers further insight into how we arrived at our results. Therefore, following your advice, we have extended the S3 table (now S4) with further details of the papers, including information about papers’ study objective and study design, and the identified main category. We, furthermore, have added a list of identified themes and categories (now S3). 

We look forward to your reply. 

Yours sincerely, also on behalf of the rest of the research team, 

Bahareh Goodarzi

---

## [Decision Letter · Decision Letter 1]

22 May 2020

Towards a better understanding of risk selection in maternal and newborn care: a systematic scoping review

PONE-D-19-34522R1

Dear Dr. Goodarzi,

We are pleased to inform you that your manuscript has been judged scientifically suitable for publication and will be formally accepted for publication once it complies with all outstanding technical requirements.

With kind regards,

Russell Kabir, PhD

Academic Editor

PLOS ONE

Additional Editor Comments (optional):

Reviewers' comments:

Reviewer's Responses to Questions

**Comments to the Author**

1. If the authors have adequately addressed your comments raised in a previous round of review and you feel that this manuscript is now acceptable for publication, you may indicate that here to bypass the “Comments to the Author” section, enter your conflict of interest statement in the “Confidential to Editor” section, and submit your "Accept" recommendation.

Reviewer #1: (No Response)

Reviewer #2: (No Response)

2. Is the manuscript technically sound, and do the data support the conclusions?

Reviewer #1: Yes

Reviewer #2: (No Response)

3. Has the statistical analysis been performed appropriately and rigorously? 

Reviewer #1: N/A

Reviewer #2: (No Response)

4. Have the authors made all data underlying the findings in their manuscript fully available?

Reviewer #1: Yes

Reviewer #2: (No Response)

5. Is the manuscript presented in an intelligible fashion and written in standard English?

Reviewer #1: Yes

Reviewer #2: (No Response)

6. Review Comments to the Author

Reviewer #1: Thank you for the opportunity to review this manuscript, which provides a comprehensive overview of current approaches to risk selection in maternity care. I did not provide the initial review of this manuscript but have reviewed carefully the authors' responses to the comments made in the initial review. These comments were relatively minor but addressing them has strengthened the manuscript. The addition of further information to (the now) table S4 provides the reader with a clearer picture of the studies reviewed and further supports the scoping review. The coding frame in S3 is also a helpful addition that adds to the transparency of the analysis.

The term "small minority" may be more accurate than "vast minority" (line 321) to describe the 14 of 210 studies that take a woman-focused perspective.

Reviewer #2: I thank authors for addressing the comments of the reviewer. I recommend to accept the paper for the publication.

7. PLOS authors have the option to publish the peer review history of their article (what does this mean?). If published, this will include your full peer review and any attached files.

Reviewer #1: No

Reviewer #2: No

---

## [Editor Report · Acceptance letter]

29 May 2020

PONE-D-19-34522R1 

Towards a better understanding of risk selection in maternal and newborn care: a systematic scoping review 

Dear Dr. Goodarzi:

I am pleased to inform you that your manuscript has been deemed suitable for publication in PLOS ONE. Congratulations! Your manuscript is now with our production department. 

With kind regards,

on behalf of

Dr. Russell Kabir 

Academic Editor

PLOS ONE